# Toward a Sheaf-Theoretic Understanding of Compositionality in Large Language Models

## Abstract

Compositionality has long been considered a fundamental aspect of human cognition - enabling the learning, manipulation, and generation of natural language. Understanding how this concept applies to Large Language Models (LLMs) and how it can be effectively evaluated remains a key challenge. In this work, we explore the potential of formalizing cognitive notions from theory, such as compositionality, to develop more nuanced evaluation frameworks for LLMs. Using a sheaf-theoretic approach, we define compositionality through four distinct conditions that capture its multifaceted nature. This formalization offers a structured perspective on evaluating LLMs, moving beyond surface-level assessments to uncover deeper insights into their behavior. Our findings suggest that theoretical frameworks like this one can play a crucial role in advancing the understanding and evaluation of LLMs, providing a foundation for more comprehensive and precise performance analyses.

## 1 Introduction

Compositionality has long been a key focus in the study of human cognition. Early work by Fodor & Pylyshyn (1988) challenged the capability of non-symbolic neural network models to be compositional due to lack of symbolic representations but Smolensky (1987),Van Gelder (1990), and Chalmers (1993) were instrumental in challenging the prevailing scepticism by asserting that the networks' intricate connection weights and activation patterns can lead to functional compositionality. However, as Aizawa & Aizawa (2003) points out, neither the symbolic nor the functional view of compositionality succeeds in building compositionality as a core tenet of the theory that can necessitate the development of compositional behaviour of a system without relying on ad-hoc assumptions. Moreover, neither the symbolic nor functional theories provide any elucidation on the processes involved in being compositional beyond a primarily concatenative lexicalist view of combining tokens or lexemes.

Such issues become more pronounced when we talk of compositionality for systems like LLMs where compositionality is not a core design feature but can emerge through the process of learning and manipulating representations. Also, LLMs today are highly performant connectionist systems and are increasingly seen as possible models of human language (Mahowald et al., 2024; Hu et al., 2024) or cognition (Kauf et al., 2023; Hardy et al., 2023; Marjieh et al., 2023; Lamprinidis, 2023) which makes it imperative for us to try and answer two important questions with respect to LLMs and their compositional abilities:

- How do we define compositionality for LLMs?
- How do LLMs perform in compositionality tasks, i.e., can these tasks help us better understand the capabilities of these models and provide insights into their overall performance?

To address the first question, we thus defer to a sheaf theoretic definition of compositionality for LLMs that uses elements of categorical compositionality (Phillips & Wilson, 2010; 2016b) and sheaf theoretic topology (Phillips, 2018; 2020) to define and delineate different aspects of compositionality. Such a way of defining compositionality has two distinct advantages: It allows us to model compositionality as a learning process that goes beyond first-order systematicity (understanding relations between entities) to the development of second-order systematicity (understanding the

structure of such relations themselves) (Phillips & Wilson, 2016a; Davis et al., 2020). Moreover, it also enables us to address compositionality *not merely* in terms of symbols – which neural networks do not explicitly possess due to polysemanticity (Huben et al., 2023; Lecomte et al.) – or through the direct composition of vectors – which is challenging due to non-linearity (Mikolov et al., 2013) – but rather in terms of patterns governing the structure of form-meaning mappings that models must learn and represent. Specifically, we model compositionality as a sheaf-theoretic phenomenon where systematic generalization capabilities arise from sheaving constructions performed on presheaves via sheaf morphisms.

Using our definition of compositionality, we formalize the possible structure of tasks needed for evaluating the different processes and aspects linked to developing compositionality. We also evaluate a wide range of LLMs on our tasks and try to determine whether performance on compositional tasks is capable of illuminating pitfalls and overall performance trends of different LLMs. Our findings reveal that the tasks are capable of reaffirming some well-known performance trends, e.g., larger models are usually better, and detecting lesser known ones, e.g., instruction-tuned models can be quite inconsistent across benchmarks. This suggests that the connection between compositionality and model performance might not be coincidental. Just as compositionality underpins human cognition, it most likely is also a fundamental characteristic of LLMs.

## 2 RELATED WORK

The investigation of compositional abilities of LLMs is not a new area of work but one of the main issues has been that most works do not adhere to a common notion of compositionality. Earlier works focused on analyzing compositional abilities in trained artificial neural networks like Lake & Baroni (2018) and Kim & Linzen (2020a) where compositionality is considered a process of uncovering the underlying syntactical structure of phrases to generalize correctly. Hupkes et al. (2020) proposes that compositionality is more than simple syntactic structure and breaks down the notion of compositionality into four aspects (systematicity, productivity, substitutivity, localism and overgeneralisation)- while this was the first work to address the complex nature of compositionality, the primary assumptions still centred around syntactic structure recovery. Moreover, these works focus on networks trained specifically for the task at hand and were before the rise of current LLMs which are highlighted by their pretraining and finetuning regimes.

For LLMs, the question of defining compositionality becomes more complex- given pretraining on a different tasks these models generalize extremely well on new tasks but how can we define or understand this compositional generalization ability in such models? Most works that investigate compositionality in LLMs adhere to the general notion of compositionality as building up of complex expressions from simple ones (Lake & Baroni, 2018; Kim & Linzen, 2020b; Hupkes et al., 2020; Lepori et al., 2023; Drozdov et al., 2022; SHAO et al., 2023; Zhou et al., 2023), none of which provide us with a formalization of the notion of compositionality and give any insights into what models need to learn to become compositional. Some recent works have considered compositionality as the ability to perform multi-hop reasoning (Dziri et al., 2024; Xu et al., 2024; Okawa et al., 2024) which is somewhat misleading as this notion of combining solutions to subproblems is far removed from the concept of compositional generalization as discussed in language and cognition sciences. Moreover, such notions of compositionality are overly symbolic and do not consider the proclivities of neural networks which are capable of a different manifestation of functional compositional abilities Smolensky (1987); Van Gelder (1990); Chalmers (1993). Defining compositionality in a symbolic or functional framework is not only limiting in terms of understanding and defining the processes that lead to compositionality, but it also restricts our interpretation of the term to learning lower order relations as opposed to higher order relations and morphisms that enable generalization in language.

In cognitive sciences, however, there has been some work in attempting a more formal understanding of compositionality that goes beyond the typical symbolic notion of compositionality Rappe (2022); Montemayor & Balci (2007) and focuses on LLM-like connectionist architectures Martin & Doumas (2020); Elmoznino et al. (2024). The most significant of such work for our purposes is the characterization of compositionality in terms of uncovering the underlying structure of data by learning the mathematical structures that characterize the data Phillips & Wilson (2010; 2016b); Phillips (2018; 2020)- such a notion of compositionality is not dependent on symbolic notions of

combining symbols to build up complex expressions and also highlights what kinds of structures models need to develop for compositional generalization, which makes this approach suitable to analyzing systems like LLMs which are not symbolic in principle.

# 3 DEFINING COMPOSITIONALITY

We adopt a sheaf-theoretic approach to compositionality for LLMs, incorporating elements of categorial compositionality and sheaf-theoretic topology to define various aspects of it. This approach offers two key benefits: it models compositionality as a learning process extending beyond first-order systematicity (relations between entities) to second-order systematicity (relations between relations) (Phillips & Wilson, 2016b;a). Additionally, it frames compositionality not merely in terms of symbols or vector composition, but as patterns in form-meaning mappings that models must learn, using sheaving constructions and morphisms to achieve systematic generalization (Phillips, 2018).

In general, a sheaf is defined in the following manner: Let $X$ be a **topological space**. A **sheaf** $\mathcal{F}$ on $X$ is a functor from the **category of open sets** $\text{Open}(X)$ to the category of sets, satisfying the following conditions:

1. For every open set $U \subseteq X$, there is a set $\mathcal{F}(U)$, called the **section** of $\mathcal{F}$ over $U$.

2. If $V \subseteq U$, then there is a restriction map $\rho_{U,V} : \mathcal{F}(U) \to \mathcal{F}(V)$.

3. **Gluing condition**: If $\{U_i\}$ is an open cover of $U$ and sections $s_i \in \mathcal{F}(U_i)$ agree on the overlaps (i.e., $s_i|_{U_i \cap U_j} = s_j|_{U_i \cap U_j}$), then there exists a unique section $s \in \mathcal{F}(U)$ such that $s|_{U_i} = s_i$ for all $i$.

4. **Locality condition:** If $s, t \in \mathcal{F}(U)$ are sections such that for each $i \in I$, $s|_{U_i} = t|_{U_i}$, then $s = t$.

Another concept from sheaf theory that facilitates the preservation of local-to-global information, is a natural transformation.

**Natural Transformation:** If $\mathcal{F}, \mathcal{G}$ are sheaves on a topological space $X$, viewed as functors from the category of open sets of $X$ (denoted by $\textbf{Open}(X)$) to the category of sets (or other suitable categories), then a natural transformation between two sheaves $\mathcal{F}$ and $\mathcal{G}$ is a family of maps:

$$\eta_U : \mathcal{F}(U) \to \mathcal{G}(U) \quad \text{for each open set } U \subseteq X,$$

such that for every inclusion of open sets $V \subseteq U$, the following diagram commutes:

$$
\begin{array}{ccc}
\mathcal{F}(U) & \xrightarrow{\text{res}^{\mathcal{F}}_{U,V}} & \mathcal{F}(V) \\
\downarrow \eta_U & & \downarrow \eta_V \\
\mathcal{G}(U) & \xrightarrow{\text{res}^{\mathcal{G}}_{U,V}} & \mathcal{G}(V)
\end{array}
$$

where $\text{res}_{U,V}$ denotes the restriction maps of the sheaves $\mathcal{F}$ and $\mathcal{G}$.

In the linguistic topological space, the property of compositional generalization can thus be understood as the structuring of sheaves from presheaves where gluing and locality conditions ensure that the local data (meanings, transformations) are consistent when combined globally, which parallels systematic compositionality in language – ensuring that local rules generalize across contexts. Moreover, being compositional in a way as to appropriately arrive at global information from local requires learning appropriate natural transformations, with commuting restrictions, for the purposes of preserving the local-global structures in an appropriate manner. Thus, for a model to be compositional, it must learn the following:

1. RESTRICTION MAPS: The ability to define proper restriction maps which ensures that data assigned to larger sets can be consistently related to smaller sets across sections.

2. GLUING CONDITIONS: The ability to avoid violations of the gluing conditions i.e. discover appropriate overlaps while discovering global sections.

3. LOCALITY CONDITIONS: The ability to avoid violations of the locality conditions i.e. determine when the local sections of data come from a global section and when they do not.

4. LEARNING NATURAL TRANSFORMATIONS:The ability to discover natural transformations that preserve the coherence of sheaves.

Now for each of the four aspects of being compositional, we define formalization of a task that can test these properties and also come up with concrete language processing tasks or datasets which we use to evaluate large language models.

## 3.1 EVALUATING RESTRICTION MAPS

Let $X$ be a topological space, and let $F$ be a sheaf over $X$. For any open set $U \subset X$, the sheaf assigns a set of sections $F(U)$ to $U$, representing data or objects over $U$.

For open sets $V \subseteq U$, there is a restriction map:

$$\text{res}_{U,V} : F(U) \rightarrow F(V),$$

which maps sections over $U$ to sections over $V$, ensuring consistency. For a section $s \in F(U)$, the restriction map ensures that:

$$\text{res}_{U,V}(s) = s_V \quad \text{where} \quad s_V \in F(V).$$

This maintains the consistency of data from larger sets to smaller sets. A violation occurs when the section on $U$ does not restrict consistently to $V$:

$$\text{res}_{U,V}(s) \neq s_V,$$

indicating that global data is inconsistent with local data. Consider open sets $U_1, U_2 \subset U$ with $U_1 \cap U_2 \neq \emptyset$. Sections $s_1 \in F(U_1)$ and $s_2 \in F(U_2)$ must agree on their overlap:

$$\text{res}_{U_1 \cap U_2, U_1}(s_1) = \text{res}_{U_1 \cap U_2, U_2}(s_2).$$

Failure to satisfy this gives:

$$\text{res}_{U_1 \cap U_2, U_1}(s_1) \neq \text{res}_{U_1 \cap U_2, U_2}(s_2) \implies s \in F(U_1 \cup U_2).$$

For $U \subset X$ covered by open sets $U_1, U_2, \ldots, U_n$, restriction maps ensure that sections $s_i \in F(U_i)$ agree on overlaps:

$$\text{res}_{U_i \cap U_j, U_i}(s_i) = \text{res}_{U_i \cap U_j, U_j}(s_j),$$

so that we can glue these sections to form a global section over $U$. A violation occurs when:

$$\text{res}_{U_i \cap U_j, U_i}(s_i) \neq \text{res}_{U_i \cap U_j, U_j}(s_j),$$

which prevents forming a consistent global section. The restriction map ensures that local and global data are consistent. Failure of the restriction map prevents gluing local sections into a global section, violating the sheaf's core properties.

The `SCAN dataset` Lake & Baroni (2018) provides an appropriate task to test the understanding of the formation of restriction maps in LLMs. It involves simple commands ("jump twice") paired with corresponding action sequences ("JUMP JUMP"). The model is expected to ensure that the mappings for complex instructions can be restricted consistently to simpler components. For instance, "jump twice" should be restricted to "jump" in a way that aligns with the learned mapping for "jump." If the model fails to consistently apply the restriction, it violates the restriction map property, indicating it cannot generalize compositionally across instructions. For more details on the suitability of this dataset for this task, please refer to A.1.

## 3.2 EVALUATING GLUING CONDITIONS

Let $X$ be a topological space and $\{U_i\}_{i \in I}$ be an open cover of $X$. For each open set $U_i$, a sheaf $F$ assigns sections (data) $s_i \in F(U_i)$. $A \in F(U_1)$ is a section defined over an open set $U_1 \subset X$ and $CA \in F(U_2)$ is a section defined over another open set $U_2 \subset X$, where $CA$ represents a compound form of $A$. Let the sets $U_1$ and $U_2$ overlap, i.e., $U_1 \cap U_2 \neq \emptyset$.

If the relation between $A$ and $CA$ is not properly determined, leading to:

$$s_1(A)|_{U_1 \cap U_2} \neq s_2(CA)|_{U_1 \cap U_2},$$

then there is no unique global section $s \in F(U_1 \cup U_2)$ that can satisfy both:

$$s|_{U_1} = s_1(A) \quad \text{and} \quad s|_{U_2} = s_2(CA).$$

Thus, the failure to determine the relation between $A$ and $CA$ constitutes a violation of the gluing condition. can be expressed as:

$$s_1(A)|_{U_1 \cap U_2} \neq s_2(CA)|_{U_1 \cap U_2} \implies s \in F(U_1 \cup U_2).$$

LLMs should be able to understand the violations of gluing condition where present. To test this in LLMs, we use our version of the AddOne Task Pavlick & Callison-Burch (2016) with the mini `Antails Dataset`. For a given sentence with a noun (N) like *The runner set a record*, we substitute N with an adjective – noun combination like *The runner set a new record* and test the model to see whether it can understand the entailment pattern. The model here has to maintain its understanding of entailment patterns with adjective substitution. Please refer to A.2 for more details on the suitability of this task for testing this condition in LLMs.

### 3.3 EVALUATING LOCALITY CONDITIONS

Let $U \subseteq X$ be a topological space and $F$ be a sheaf on $U$, assigning sections $s_i \in F(U_i)$ to open sets $U_i \subset U$. Consider a task where we are given a triple $(a, b, c)$, where $a$ and $b$ are semantically related, but $a$ and $c$ are not. $s_{ab}$ is the section over an open set $U_1 \subset U$, capturing the semantic relationship between $a$ and $b$, $s_{ac}$ is the section over an open set $U_2 \subset U$, capturing the semantic relationship between $a$ and $c$. $U_1 \cap U_2 \neq \emptyset$ represents the overlap between the regions covered by $s_{ab}$ and $s_{ac}$.

If the sections $s_{ab}$ and $s_{ac}$ were to satisfy the locality condition, we would require:

$$s_{ab}|_{U_1 \cap U_2} = s_{ac}|_{U_1 \cap U_2}$$

However, since $a$ and $c$ are not semantically related, the sections $s_{ab}$ and $s_{ac}$ should differ in the overlap $U_1 \cap U_2$. If the model fails to distinguish between $s_{ab}$ and $s_{ac}$, this would violate the locality condition because it would incorrectly equate the unrelated pair $(a, c)$ with the related pair $(a, b)$, implying:

$$s_{ab}|_{U_1 \cap U_2} = s_{ac}|_{U_1 \cap U_2} \quad \text{(incorrect, as } a \text{ and } c \text{ are not related)}$$

This failure results in: $s_{ab} = s_{ac}$ which is a contradiction, since:

$$s_{ab} \neq s_{ac} \quad \text{(as } a \text{ and } b \text{ are semantically related, but } a \text{ and } c \text{ are not)}.$$

Thus, this failure to distinguish between $(a, b)$ and $(a, c)$ constitutes a violation of the locality condition in sheaf theory.

To evaluate LLMs on their ability to respect locality conditions, we propose the `COMPCOMB dataset`- a new task type using a handcrafted toy dataset which is a novel contribution of this work (more details on suitability of dataset for this task in A.3). Each data point consists of a triple – a noun, an adjective that goes with the noun, and an exocentric compound which contains the noun. For example, (coat, trenchcoat and turncoat) – when we take the word *"coat"*, we know that *"trenchcoat"* ( a special type of coat) is closely related to it but the exocentric compound *"turncoat"* (a betrayer) is not since it is semantically different. This tests the LLM's ability to distinguish between genuine compounds and combinations by avoiding generalization on the basis of surface forms.

## 3.4 LEARNING UNIVERSAL TRANSFORMATIONS

Let $F_A$, $F_B$, and $F_C$ be sheaves over a topological space $X$. We are given the following mappings:

$$\phi_{A,B} : F_A \to F_B,$$

$$\phi_{A,C} : F_A \to F_C.$$

The task is to find a mapping:$\phi_{A,BC} : F_A \to F_{BC}$ where $F_{BC}$ represents a combined sheaf constructed from $F_B$ and $F_C$. The sheaf $F_{BC}$ combines the data from $F_B$ and $F_C$ in a way that respects both the mappings $\phi_{A,B}$ and $\phi_{A,C}$. A natural transformation $\eta$ must respect the restriction maps of the sheaves. If the task of finding $\phi_{A,BC} : F_A \to F_{BC}$ fails, this indicates that we cannot construct a natural transformation between the sheaves $F_A$ and $F_{BC}$. Specifically, the failure occurs if the mappings $\phi_{A,B}$ and $\phi_{A,C}$ are inconsistent with the desired mapping $\phi_{A,BC}$. This would result in the failure of the following commutative diagram:

$$
\begin{array}{ccc}
F_A & \xrightarrow{\phi_{A,BC}} & F_{BC} \\
\downarrow \phi_{A,B} & & \downarrow \\
F_B & & F_C
\end{array}
$$

If $\phi_{A,B}$ and $\phi_{A,C}$ do not align in a way that allows the construction of $\phi_{A,BC}$, then there is no natural transformation between $F_A$ and $F_{BC}$, indicating a failure to establish the relationship between $A$, $B$, and $C$. This indicates that the failure to relate $F_A \to F_{BC}$ stems from the inconsistency between $\phi_{A,B}$ and $\phi_{A,C}$, violating the conditions required for a natural transformation between the sheaves.

An LLM must be able to distinguish appropriately when the diagram commutes and when it doesn't i.e. between situations when the natural transformation exists and when it doesn't. To test this in LLMs, we use the `PLANE Dataset` Bertolini et al. (2022) that tests adjective – noun entailment in a situation where the entailment pattern for an AN – N and AN – H (where AN is the adjective – noun combination, N is the noun and H is a hypernym of N) combination is already given and the model is tested on entailment of AN – AH combination. Please refer to A.4 for more details on the suitability of this task for testing this condition in LLMs.

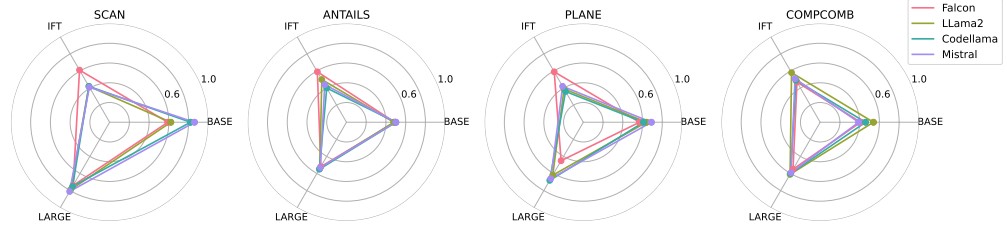

Figure 1: Radar plots comparing the accuracy of four models (Falcon, Llama, Codellama, Mistral) across four datasets (SCAN, ANTAILS, PLANE, COMPCOMB) in the Log Probabilities setup. Each plot shows the performance of the models for three types (BASE, IFT, LARGE). The radial axis represents accuracy, scaled from 0 to 1.

## 4 EXPERIMENTS

### 4.1 MODELS

To evaluate compositionality across Large Language Models (LLMs), we selected four distinct model families: `Falcon` (Almazrouei et al., 2023), `Llama2` (Touvron et al., 2023), `Codellama` (Roziere et al., 2023), and `Mistral`(Jiang et al., 2023). Each model family represents state-of-the-art LLM architectures, making them suitable for analyzing compositional behaviour.

For each model family, we selected three models for testing:

- Base Model (Base): A 7 billion parameter model that serves as the foundational version of each family.

- Instruction – Finetuned Model (IFT): The same 7B base model, further fine-tuned with instruction-tuning to enhance task performance.
- Scaled Model (Large): A model variant with a higher parameter count, ranging from 13B to 70B, depending on availability within each family. These larger models allow us to investigate how scaling affects compositional behavior.

The diversity in models ensures that our analysis captures how both model complexity and tuning approaches impact compositionality. Refer to B.1 for more details on the models used.

## 4.2 EXPERIMENTAL SETUP

The four tasks and datasets utilized in this work can be broadly categorized into two distinct types: behavioural and representational. This classification is based on the nature of the evaluation employed for each dataset.

**Behavioural Analysis**: These datasets evaluate the model based on its input – output behaviour, i.e., the focus is on how the model behaves when presented with specific tasks or queries. The behavioural datasets include:

- The SCAN Dataset, which tests a model's ability to generalize simple instruction patterns to more complex ones. We use 100 samples from the SCAN dataset.
- The Antails Dataset, which focuses on distinguishing between related and unrelated noun – adjective – exocentric compound combinations. We adapt 70 samples from the original AddOne dataset Pavlick & Callison-Burch (2016) and use it for our evaluation.
- The PLANE Dataset, which evaluates the model's understanding of entailment relations between adjective –noun pairs and their hypernyms. The PLANE dataset contains five train-test splits and we use one test split consisting of 1500 samples.

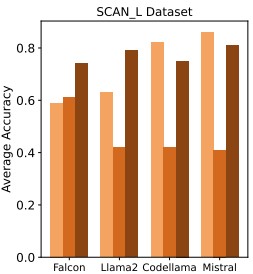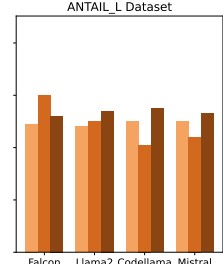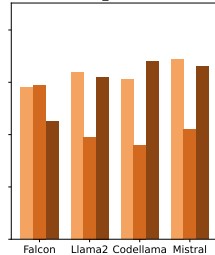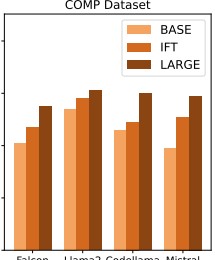

Figure 2: Comparison of average across different model families (Falcon, Llama2, Codellama, Mistral) and model types (BASE, IFT, LARGE) for four datasets (SCAN, Antails, Plane, CompComb). Each bar represents the average accuracy across 2 prompt variations.

Each of these behavioural datasets is evaluated with a comparative log probability setup. The evaluation involves computing the model's log probabilities for two possible completions: one being the correct option and the other the control (incorrect option). The model's preference between the two completions is determined by comparing their log probabilities and the setup focuses on the model's probabilistic confidence in its outputs. The completion with the higher log probability is considered indicative of the model's judgement and we conduct experiments with two prompts to ensure robustness for our results. For both the `Antails Dataset` and the `PLANE Dataset`, which involve binary classification tasks, the two completions correspond to entailment and non-entailment outcomes.

The prompt completions used in our evaluation are simple prompts. We choose not to use advanced prompts like few-shot Wei et al. (2021) and chain of thought Wei et al. (2022c) to avoid giving undue advantages to the instruct models since they are typically trained to show the best performance with advanced instruction prompts Longpre et al. (2023). Moreover, we also choose the log probability evaluation instead of prompt-output evaluation due to problems with prompted output

evaluation. Recent research indicates that prompt outputs of LLMs are often misleading (Sclar et al., 2024; Turpin et al., 2024; McCoy et al., 2023) with log-likelihood comparisons being better for understanding model competence on most tasks (Hu & Levy, 2023; Kauf et al., 2024), and we find similar uncertainties and high variation across very similar prompts in prompting output evaluations for our task (refer B.3 for more details), so we adopt the log probability setup for conducting our evaluations.

**Representational Analysis**: This dataset type evaluates the model based on its internal representations, rather than its input – output behaviour. The `Compcomb Dataset` is specifically designed to examine how well the model's internal representations encode the relationships between related and unrelated adjective – noun and exocentric compound pairs. It is a dataset with 50 samples.

To evaluate the model's representations, we extract data from two key layers of the model:

- The embedding layer: This layer captures the model's initial word representations before any processing from the deeper layers.
- The final hidden layer: This layer captures the model's most complex and abstracted representations, which reflect its deep understanding of the input after all layers have processed it.

For each layer, we get representations of the model for each word in the triple and the model is considered to be accurate if its representations for noun and adjective – noun combinations are closer than the noun and semantically unrelated compound representations. By comparing the model's representations in these two layers, we can gain insights into how well the model captures semantic relationships and distinctions between input items (such as distinguishing between a noun and its related and unrelated compounds). This setup allows for an analysis of the model's ability to differentiate semantically related pairs from unrelated ones based purely on internal representation quality.

Table 1: Results from our evaluation setup across 4 datasets and 4 model families comparing a base model (7b), an instruction-tuned model (IFT) and a large model (above 7b). The variations recorded are across two prompts in the setup. There are no variations for COMPCOMB since it is based on analysing representations.

(a) SCAN

| Model | BASE | IFT | LARGE |
|---|---|---|---|
| Falcon | 0.59±0.02 | 0.61±0.01 | 0.74±0.03 |
| Llama 2 | 0.63±0.01 | 0.42±0.02 | 0.79±0.01 |
| Codellama | 0.82±0.05 | 0.42±0.03 | 0.75±0.00 |
| Mistral | 0.86±0.00 | 0.41±0.02 | 0.81±0.05 |

(b) ANTAILS

| Model | BASE | IFT | LARGE |
|---|---|---|---|
| Falcon | 0.50±0.01 | 0.59±0.05 | 0.52±0.02 |
| Llama 2 | 0.48±0.02 | 0.50±0.00 | 0.54±0.03 |
| Codellama | 0.50±0.01 | 0.41±0.06 | 0.55±0.02 |
| Mistral | 0.50±0.03 | 0.44±0.07 | 0.53±0.06 |

(c) PLANE

| Model | BASE | IFT | LARGE |
|---|---|---|---|
| Falcon | 0.58±0.03 | 0.59±0.05 | 0.45±0.14 |
| Llama 2 | 0.64±0.02 | 0.39±0.04 | 0.62±0.01 |
| Codellama | 0.61±0.04 | 0.36±0.15 | 0.68±0.02 |
| Mistral | 0.69±0.00 | 0.42±0.25 | 0.66±0.03 |

(d) COMPCOMB

| Model | BASE | IFT | LARGE |
|---|---|---|---|
| Falcon | 0.41 | 0.47 | 0.55 |
| Llama 2 | 0.54 | 0.58 | 0.61 |
| Codellama | 0.46 | 0.49 | 0.60 |
| Mistral | 0.39 | 0.51 | 0.59 |

### 4.3 RESULTS AND ANALYSIS

Our experiments evaluated compositionality in terms of learning different aspects of creating a sheaf that leads to complete compositional generalization in a model. In 1 and 2 we compare the performances of each model type (base, instruction following checkpoint, and larger model) where each subplot indicates the results for a dataset/condition and in 1 we provide the actual accuracies of model performance across each dataset.

Across the four model families tested, we present a brief overview of how they perform on each aspect of compositionality:

**Restriction Condition**: For the `SCAN Dataset`, which tests the restriction conditions, we observe in that while none of the models perfectly satisfy the restriction condition, within each model family

the largest models get the highest accuracies showing an improved understanding in this aspect of compositionality. This aligns with most LLM evaluation studies on the impacts of scaling (Wei et al., 2022a; Ouyang et al., 2022; Chung et al., 2024). However, more surprisingly, we see that instruction tuned models perform the worst for `Llama2`, `Codellama`, and `Mistral` – indicating that instruction tuning likely leads to a loss in the development of restriction maps which could be explained by the fact that while the model retains it's most important generalizations, it loses some local information to accommodate instruction tuning, leading to loss of restriction mapping. This also echoes more recent research that investigates the negative impacts and knowledge degradation of instruction tuned or aligned models (Ghosh et al., 2024; Sun et al., 2024).

**Gluing Condition**: The evaluation of the gluing condition with the `Antails Dataset` shows a more variable pattern of behaviour across model families – while larger models are better for the majority of model families, instruction tuning leads to better performance in `Falcon` and `Llama2` while it leads to worse performance in the acquisition of gluing condition for both `Codellama` and `Mistral` models. Such a variance across model types and families might be indicative of a higher level of difficulty in acquiring the gluing conditions of compositionality, making it very specific to different model training data and procedures.

**Locality Condition**: We evaluate the locality condition with our `Compcomb Dataset` and observe more stable trends across all families of models (`Falcon`, `Llama2`,`Codellama`, and `Mistral`) showing that instruction tuned models do better than base models while scaled models still perform the best. This indicates that instruction tuning and scaling both contribute to improved learning of the locality conditions and the learning process might be more stable across models, as compared to the gluing condition. Compared with the restriction condition, we see that while instruction tuning leads to loss of information on local sections of the topology and the ability to distinguish when the global sections can be reconstructed and when they cannot, it still systematically retains information on the presence of a unique global section.

**Natural Transformation**: The `PLANE Dataset` is targeted at analysing the ability of models to find the appropriate conditions for natural transformations between sheaves. The performance trends here are more stable across model families where the larger models show uniform improvements in their abilities to realize natural transformations inherent in the data. Also, models in the `Llama2`, `Codellama` and `Mistral` family show similar patterns of learning as the restriction condition where instruction tuned models show worsening abilities in recognizing the correct natural transformation. Another interesting pattern emerges here- exactly the same model families where instruction tuning harmed learning of the gluing condition also shows inverse scaling (Wei et al., 2022b; Michaelov & Bergen, 2022; McKenzie et al., 2023; Gupta, 2023) for learning of natural transformations. This might be indicative of a subtly stronger interplay between learning restrictions and finding natural transformations that gets reflected in the compositional abilities of the model.

## 5 DISCUSSION

Our work focuses on the development of a sheaf-theoretic interpretation of compositionality that portrays compositional generalization as emerging from the ability to construct sheaves and natural transformations between sheaves. Such an interpretation is not only advantageous from a cognitive point of view, where it has been found to be relevant for understanding reasoning processes and pitfalls in humans (Phillips, 2018) but also from the point of view of understanding and evaluating capabilities of models of language like LLMs.

- **Systematic Understanding of Compositionality:** By breaking down the complex phenomenon of compositionality into four testable conditions related to constructing proper sheaves and morphisms, our approach allows for precise evaluation of this phenomenon in models. These conditions provide the foundation for targeted understanding of specific aspects of compositionality, enabling a more structured and systematic evaluation framework for LLMs. It allows us to break down the complex phenomenon of compositionality into four aspects of building a proper sheaf/sheaf morphism.

- **Nuanced Task-based Evaluation:** We provide a suitable task paired with four different conditions, which makes it easier to evaluate the compositional abilities of language models and analyse their performance in terms of each aspect. Our testable conditions allow

us to identify four tasks that map to each condition and the focus here is to show that formalization should lead to testable conditions not to estbalish that the tasks we show are the only or optimal tests of compositionality.

- **Potential Downstream Applications:** Compositionality has been considered a core feature of human language abilities which leads to their superior performance in tasks like reasoning, generalization and quick learning from limited data. As models of language, we can also expect that compositionality might be a core feature driving downstream performance of models. The performance of LLMs in this small set of tasks already reveals different behavioural trends that have been observed from different tasks and benchmarks-both scaling and inverse scaling but also both improvement and worsening performance of fine tuned models. This indicates that the aspects of compositionality delineated here might have a causal impact on general reasoning capabilities in models and might even be indicative of their overall performance trends.

- **Dynamic View of Compositionality:** The view of compositionality as a dynamic process (instead of an ideal static arrangement of discrete symbols) is more amenable to interpretability. By focusing on how local connections and transformations aggregate to form global representations, we can analyse the development of different aspects of compositionality in different model components to gain a clearer insight into the inner workings of models, allowing us to identify how individual parts contribute to the whole. This, in turn, can facilitate the debugging, refining, and optimizing of models by targeting specific local processes that influence overall performance and consistency in such models.

In summary, our approach to compositionality offers a comprehensive framework that enriches both cognitive and computational understanding of how complex structures are formed from simpler components and enables a more structured evaluation of their reasoning abilities. This work is not aimed at finding the best definition of compositionality or the ideal set of tasks to measure compositionality in LLMs, but rather it aims to highlight that our current understanding of compositionality-especially for connectionist systems like LLMs- is quite limited and that ultimately, this perspective not only advances the theoretical understanding of compositionality but can also provide practical tools for evaluating and improving the performance of complex systems like language models.

## 6    LIMITATIONS & FUTURE WORK

Our work is aimed at attempting a formal definition of compositionality, influenced by theories from human cognition, and providing possible tasks that could be used to test LLMs under such formal frameworks- however, we do not claim that our framework is the only one or even that the tasks we choose to assess compositionality are the best- merely that compositionality is a complex phenomenon that deserves a more nuanced formal definition in case of LLMs and that such formalization can also help us choose tasks for better insightful evaluation in such models. We leave it up to future works to develop similar formal notions of compositionality and develop more nuanced evaluations for the same.

In terms of datasets and models, our collection is small i.e we use small dataset samples and few models due to compute limitations. Moreover, some of our datasets are limited in size and they may not be the perfect ones to capture each facet of compositionality and further research should focus on large scale evaluation with larger datasets and developing even better datasets suited to testing each condition in the framework.

The link between compositionality and overall model performance is suggested but not fully established. It remains uncertain to what extent compositionality directly impacts general model capabilities or whether other factors like model size or training data play a larger role.

An area of future work is the generalization and application of this framework to a wider range of models. Currently, our work focuses on specific LLM types such as instruction tuned and scaled models due to current compute limitations. However, it could be used to evaluate models with a wider range of sizes and training or finetuning methods to explore how different processes of learning can impact compositionality in models. Moreover, the framework is general enough to allow potential generalization to test composition and reasoning abilities in different types of emerging language model architectures (Fu et al., 2023; Hasani et al., 2023).

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

# A    APPENDIX A

## A.1    SCAN FOR RESTRICTION CONDITION

In sheaf theory, for a topological space $X$ and an open set $U \subset X$, a sheaf $F$ assigns to $U$ a set of sections $F(U)$, representing data or mappings over $U$. If $V \subset U$, the restriction map $\mathrm{res}_{U,V} : F(U) \to F(V)$ ensures that the data on $V$ is the restriction of the data on $U$.

For a section $s \in F(U)$, the restriction to the subset $V$ is:

$$\mathrm{res}_{U,V}(s) = s|_V,$$

which guarantees that the local data $F(V)$ is consistent with the global data $F(U)$.

The SCAN task consists of simple instructions ("turn left twice") paired with target outputs("LTURN LTURN"). Let $X$ represent the set of all possible instructions, and let $F$ be a sheaf that assigns to each open set $U \subset X$ the corresponding action mappings for the instructions in $U$. For instance:

$$F(U_{\mathrm{simple}}) = \{\text{action mappings for simple instructions}\},$$
$$F(U_{\mathrm{complex}}) = \{\text{action mappings for complex instructions}\}.$$

For a complex instruction $U_{\mathrm{complex}}$ and a subset $U_{\mathrm{subcomplex}} \subset U_{\mathrm{complex}}$, the restriction condition requires that the action mapping for the complex instruction $s_{\mathrm{complex}} \in F(U_{\mathrm{complex}})$ restricts consistently to the simpler instruction in $U_{\mathrm{subcomplex}}$. This is expressed as:

$$\mathrm{res}_{U_{\mathrm{complex}},U_{\mathrm{subcomplex}}}(s_{\mathrm{complex}}) = s_{\mathrm{subcomplex}}.$$

A violation occurs when the learned mapping for the complex instruction does not restrict consistently to its subcomponents. Mathematically, this violation can be represented as:

$$\mathrm{res}_{U_{\mathrm{complex}},U_{\mathrm{subcomplex}}}(s_{\mathrm{complex}}) \neq s_{\mathrm{subcomplex}}.$$

This failure indicates that the model's mapping for the complex instruction does not align with its simpler parts, which would violate the **restriction map** property in sheaf theory. Let us look at a specific example:

Let $U_{\mathrm{jump}}$ represent the instruction "jump" and $U_{\mathrm{jump\ twice}}$ represent the instruction "jump twice." The restriction condition requires that the mapping for the complex instruction "jump twice" reduces to the simpler instruction "jump":

$$\mathrm{res}_{U_{\mathrm{jump\ twice}},U_{\mathrm{jump}}}(s_{\mathrm{jump\ twice}}) = s_{\mathrm{jump}}.$$

A failure occurs when:

$$\mathrm{res}_{U_{\mathrm{jump\ twice}},U_{\mathrm{jump}}}(s_{\mathrm{jump\ twice}}) \neq s_{\mathrm{jump}},$$

indicating that the model fails to restrict the mapping for the complex instruction correctly to the simpler one. For any instruction $\alpha$ composed of subinstructions $\beta$ and $\gamma$, the restriction conditions require:

$$\mathrm{res}_{U_\alpha,U_\beta}(s_\alpha) = s_\beta, \quad \text{and} \quad \mathrm{res}_{U_\alpha,U_\gamma}(s_\alpha) = s_\gamma.$$

A violation occurs when:

$$\mathrm{res}_{U_\alpha,U_\beta}(s_\alpha) \neq s_\beta \quad \text{or} \quad \mathrm{res}_{U_\alpha,U_\gamma}(s_\alpha) \neq s_\gamma.$$

This shows that the model's understanding of the complex instruction $\alpha$ does not correctly restrict to its components $\beta$ or $\gamma$, violating the sheaf's restriction requirement. Thus, the SCAN task tests the restriction map property in sheaf theory.

## A.2    ANTAILS FOR GLUING CONDITION

The gluing condition ensures that if sections over different open sets agree on their overlaps, they can be combined to form a global section over the union of those sets. In the context of LLMs, understanding how well the model glues together local information to form a correct global interpretation is crucial. The Antails task naturally emerges as an ideal test for this, as it evaluates whether the model can combine information from local contexts (substituting a noun with an adjective-noun combination) into a global sentence-level entailment. For a given sentence with a noun (N) like

"The runner set a record", we substitute N with an adjective – noun combination like "The runner set a new record" and test the model to see whether it can understand the entailment pattern. The model here has to maintain it's understanding of entailment patterns with adjective substitution.

It tests whether a model can identify violations of the gluing condition by evaluating its ability to combine local modifications in a sentence into a globally consistent interpretation. Specifically, the task examines whether the model can recognize whether the entailment patterns between a sentence and its modified version remain consistent after a substitution.

Let $X$ be a topological space representing the set of all sentences. Consider two open sets $U_1 \subset X$ and $U_2 \subset X$ corresponding to two different forms of the same sentence: - $U_1$ contains the original sentence with a noun $N$, - $U_2$ contains the sentence with an adjective-noun compound $CA$ replacing $N$.

Let:
$$A \in F(U_1) \quad \text{and} \quad CA \in F(U_2)$$
represent the sections (data) corresponding to the original sentence $A$ and the modified sentence $CA$, respectively.

The gluing condition requires that if the sections $A$ and $CA$ agree on the overlap $U_1 \cap U_2$, i.e.,
$$s_1(A)|_{U_1 \cap U_2} = s_2(CA)|_{U_1 \cap U_2},$$
then there exists a global section $s \in F(U_1 \cup U_2)$ such that:
$$s|_{U_1} = s_1(A) \quad \text{and} \quad s|_{U_2} = s_2(CA).$$

The task examines whether the model can combine the local information from $A$ and $CA$ into a globally consistent interpretation. Specifically, the model is tasked with determining whether the global entailment pattern is preserved after the substitution of $N$ with $CA$.

For example: Let $A$ correspond to the sentence: $A$ : The runner set a record. and let $CA$ correspond to the sentence: $CA$ : The runner set a new record. The model must determine whether the global entailment of $A$ and $CA$ remains consistent. If the model can correctly identify that the entailment patterns agree, it satisfies the gluing condition. Otherwise, a failure to recognize the correct global entailment pattern indicates a violation of the gluing condition.

Mathematically, if the model fails to glue the local information, we observe:
$$s_1(A)|_{U_1 \cap U_2} \neq s_2(CA)|_{U_1 \cap U_2},$$
which implies that:
$$s \in F(U_1 \cup U_2) \quad \text{such that} \quad s|_{U_1} = s_1(A) \quad \text{and} \quad s|_{U_2} = s_2(CA).$$

Thus, the task serves as a direct test of the gluing condition, by evaluating whether the model can combine local changes (substituting $N$ with $CA$) into a coherent global interpretation of the sentence's entailment pattern.

### A.3 COMPCOMB FOR LOCALITY CONDITION

In sheaf theory, the locality condition ensures that if local sections (data) agree on overlapping regions, they must arise from the same global section. The Compcomb Dataset is designed to test whether a model can distinguish between semantically related pairs (coat and trenchcoat) and unrelated pairs (coat and turncoat) , ensuring that the model does not overgeneralize by incorrectly equating unrelated elements. This naturally aligns with the locality condition, as the task tests whether the model can correctly handle cases where local sections should differ based on semantic distinctions.

Let $U \subseteq X$ be a topological space, and let $F$ be a sheaf on $U$, assigning sections $s_i \in F(U_i)$ to open sets $U_i \subset U$. Consider a task where we are given a triple $(a, b, c)$, where $a$ and $b$ are semantically related, but $a$ and $c$ are not. $s_{ab} \in F(U_1)$ captures the semantic relationship between $a$ and $b$, while $s_{ac} \in F(U_2)$ captures the semantic relationship between $a$ and $c$, where $U_1 \cap U_2 \neq \emptyset$.

The locality condition requires that if sections agree on overlaps, they come from the same global section:

$$s_{ab}|_{U_1 \cap U_2} = s_{ac}|_{U_1 \cap U_2}.$$

However, since $a$ and $c$ are not semantically related, the sections $s_{ab}$ and $s_{ac}$ should differ on $U_1 \cap U_2$.

If the model fails to distinguish between $s_{ab}$ and $s_{ac}$, this results in:

$$s_{ab}|_{U_1 \cap U_2} = s_{ac}|_{U_1 \cap U_2} \quad \text{(incorrect)},$$

which violates the locality condition, implying:

$$s_{ab} = s_{ac} \quad \text{(contradictory, as $a$ and $b$ are related, but $a$ and $c$ are not).}$$

The Compcomb dataset is designed to evaluate whether models can respect the locality condition by avoiding overgeneralization. For each data point, we define a noun $a$ (e.g., "coat"), an adjective-noun combination $b$ that is semantically related to $a$ (e.g., "trenchcoat"), and an exocentric compound $c$ that contains $a$ but is semantically unrelated (e.g., "turncoat"). Let $s_{ab} \in F(U_1)$ represent the section capturing the semantic relationship between $a$ and $b$, and let $s_{ac} \in F(U_2)$ represent the section capturing the relationship between $a$ and $c$, where $U_1 \cap U_2 \neq \emptyset$. The model should be able to distinguish between these sections, satisfying:

$$s_{ab} \neq s_{ac}.$$

The model is tested on whether it can differentiate between these semantically related and unrelated pairs. A model failure occurs if it incorrectly generalizes the relationship between $a$ and $c$ based on surface forms, treating it as semantically similar to the relationship between $a$ and $b$. This can be formalized as:

$$s_{ab}|_{U_1 \cap U_2} = s_{ac}|_{U_1 \cap U_2}.$$

Such an equation would imply that the model overgeneralizes by equating the unrelated pair $(a, c)$ with the related pair $(a, b)$, thereby violating the locality condition. The correct behavior, respecting the locality condition, requires:

$$s_{ab}|_{U_1 \cap U_2} \neq s_{ac}|_{U_1 \cap U_2}.$$

Thus, the failure to distinguish between $(a, b)$ and $(a, c)$ constitutes a violation of the locality condition, where the model wrongly generalizes the semantic relation between unrelated elements based on surface similarity.

### A.4 PLANE FOR NATURAL TRANSFORMATIONS

In sheaf theory, a natural transformation between two sheaves ensures that mappings between objects are consistent across different spaces, respecting the relationships between the mappings. The PLANE dataset tests this ability by requiring the model to combine mappings for adjective – noun (AN – Noun) and adjective – hypernym (AN – Hypernym) pairs into a consistent, global mapping for AN – AH (adjective – hypernym combinations). If the model fails to maintain the consistency required for a natural transformation, it indicates an inability to generalize the relationships between these mappings, which the PLANE dataset is specifically designed to detect.

The PLANE Dataset evaluates whether models can construct the correct natural transformation when combining adjective – noun (AN) entailments with their hypernyms. Specifically: $\phi_{A,B}$ corresponds to the entailment mapping for the AN –Noun combination, while $\phi_{A,C}$ corresponds to the entailment mapping for the AN – Hypernym combination. The task is to find $\phi_{A,BC}$, which corresponds to the combined entailment mapping for the AN – Hypernym combination (AN – AH). For example, AN phrases containing intersective (I) adjectives (e.g., red, dead, and Finnish) describe a subset of entities subsumed by the noun itself and also a subset of entities which all have that adjective as a property. For example, a red car is both a car and a red thing. Thus, AN phrases containing intersective adjectives satisfy all forms of inference types (IT):

$$\text{red car} \models \text{car} \quad \text{(IT 1)}, \quad \text{red car} \models \text{vehicle} \quad \text{(IT 2)}, \quad \text{red car} \models \text{red vehicle} \quad \text{(IT 3)}.$$

Subsective adjectives (small, intelligent, strong etc) only satisfy IT1 and IT2 while intensional adjectives (fake, former, possible etc) only satisfy IT3.

The dataset requires the model to: 1. Understand the relationship between $\phi_{A,B}$ (AN – Noun) and $\phi_{A,C}$ (AN – Hypernym). 2. Combine these two mappings systematically to form $\phi_{A,BC}$ (AN – AH), which must respect both the AN – N and AN – H mappings.

If the model fails to construct $\phi_{A,BC}$ correctly, it demonstrates that the model cannot construct a natural transformation between these entailments. The dataset requires that the commutative diagram holds:

$$
\begin{array}{ccc}
F_A & \xrightarrow{\phi_{A,BC}} & F_{BC} \\
\downarrow \phi_{A,B} & & \downarrow \\
F_B & & F_C
\end{array}
$$

The model must ensure that the entailment patterns respect the relationships between the mappings. A failure occurs when:

$$\phi_{A,B} \quad \text{and} \quad \phi_{A,C} \quad \text{are inconsistent, leading to no valid} \quad \phi_{A,BC}.$$

Thus, the model fails to construct a natural transformation and does not properly generalize the entailment pattern from the AN – Noun and AN – Hypernym combinations to the AN – AH combination.

It is ideal for testing the model's ability to construct natural transformations. It requires the model to combine multiple mappings (AN – N and AN – H entailments) and ensure consistency when moving to the combined entailment pattern (AN – AH). If the model cannot ensure the commutative diagram holds or fails to combine the mappings, it indicates a failure in learning the natural transformation between these entailment patterns.

The Plane dataset was created by Bertolini et al. (2022) to test compositionality in language models and inference with phrase-level adjective-noun entailment. There are three different adjective classes in this dataset: intersective (I), subsective (S), and intensional (O).

The intersective adjectives (I) describe entities that can be categorized both by the noun and the adjective. For example, a "red car" is both a car and a red object. This satisfies all forms of inference. For example, $Redcar \models car$ and $Redcar \models vehicle$ (hypernym of "car") and $Redcar \models redvehicle$.

The subsective adjectives (S) describe entities that are part of the noun's category but do not necessarily share the property of the adjective. For example, a "small elephant" is an elephant but not necessarily a small entity in general. (e.g., $smallelephant \models elephant$; $smallelephant \models animal$), but not ($smallelephant \not\models smallanimal$).

The intensional adjectives (O) negate core properties of the noun. For example, a "fake gun" is not a real gun, so the first two types of inferences do not hold ($fakegun \not\models gun$; $fakegun \not\models weapon$). However, the third inference holds (e.g., $fakeGlock \models fakegun \models fakeweapon$), as the modification leads to a new subset of entities described by the hypernym of the noun.

In the main paper we present results averaged across these three categories. The different performance for every adjective class averaged across prompts and setups, is shown in Figure 3.

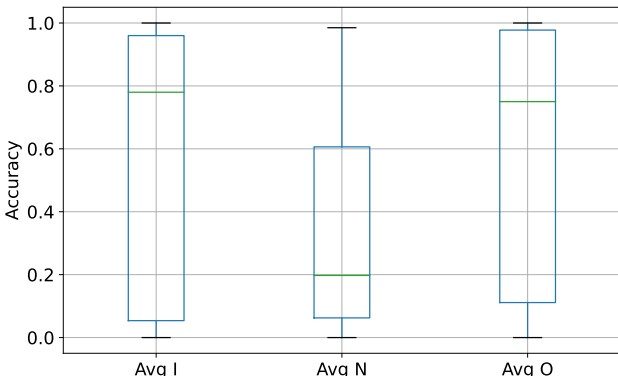

Figure 3: Average accuracy across prompts and setups for the three different adjective classes in this dataset: intersective (I), subsective (S), and intensional (O).

## B   APPENDIX B

### B.1   MODEL DETAILS

Table 2: Models used and corresponding Huggingface Hub Links

| MODEL NAME | MODEL LINK |
|---|---|
| FALCON-7B | HTTPS://HUGGINGFACE.CO/TIIUAE/FALCON-7B |
| FALCON-7B-INSTRUCT | HTTPS://HUGGINGFACE.CO/TIIUAE/FALCON-7B-INSTRUCT |
| FALCON-40B | HTTPS://HUGGINGFACE.CO/TIIUAE/FALCON-40B |
| LLAMA-2-7B-HF | HTTPS://HUGGINGFACE.CO/META-LLAMA/LLAMA-2-7B-HF |
| LLAMA-2-7B-CHAT-HF | HTTPS://HUGGINGFACE.CO/META-LLAMA/LLAMA-2-7B-CHAT-HF |
| LLAMA-2-13B-HF | HTTPS://HUGGINGFACE.CO/META-LLAMA/LLAMA-2-13B-HF |
| CODELLAMA-7B-HF | HTTPS://HUGGINGFACE.CO/CODELLAMA/CODELLAMA-7B-HF |
| CODELLAMA-7B-INSTRUCT-HF | HTTPS://HUGGINGFACE.CO/CODELLAMA/CODELLAMA-7B-INSTRUCT-HF |
| CODELLAMA-13B-HF | HTTPS://HUGGINGFACE.CO/CODELLAMA/CODELLAMA-13B-HF |
| MISTRAL-7B-V0.1 | HTTPS://HUGGINGFACE.CO/MISTRALAI/MISTRAL-7B-V0.1 |
| MISTRAL-7B-INSTRUCT-V0.1 | HTTPS://HUGGINGFACE.CO/MISTRALAI/MISTRAL-7B-INSTRUCT-V0.1 |
| MIXTRAL-8X7B-V0.1 | HTTPS://HUGGINGFACE.CO/MISTRALAI/MIXTRAL-8X7B-V0.1 |

### B.2   EVALUATION SETUP DETAILS

We use an evaluation setup to extract the log probabilities where Setup 1 and Setup 2 use different input prompts on which log probabilities are evaluated. 3 shows setup for SCAN, 4 shows setup for Antails, and 5 shows setup for PLANE.

### B.3   PROMPTING SETUP RESULTS

Here we provide results from prompting the models and evaluating their generated outputs of which option they deem more suitable in the prompt where one option was correct and the other an incorrect option . Since the model outputs were very sensitive to the different prompts and biased towards predicting specific options and selections we decided to enlist in the Appendix, but not include the results in the main paper.

Table 6: Results from the prompt setup across 4 datasets and 4 model families comparing a base model (7b), an instruction tuned model (IFT) and a large model (above 7b).

(a) SCAN

| Model | BASE | IFT | LARGE |
|---|---|---|---|
| Falcon | 0.70±0.29 | 0.36±0.26 | 0.98±0.02 |
| Llama 2 | 1.00±0.00 | 0.00±0.00 | 1.00±0.00 |
| Codellama | 0.75±0.25 | 0.00±0.00 | 1.00±0.00 |
| Mistral | 1.00±0.00 | 0.00±0.00 | 0.98±0.02 |

(b) ANTAILS

| Model | BASE | IFT | LARGE |
|---|---|---|---|
| Falcon | 0.51±0.00 | 0.47±0.00 | 0.54±0.02 |
| Llama 2 | 0.51±0.01 | 0.50±0.00 | 0.59±0.01 |
| Codellama | 0.50±0.00 | 0.00±0.00 | 0.53±0.03 |
| Mistral | 0.50±0.00 | 0.41±0.13 | 0.52±0.02 |

(c) PLANE

| Model | BASE | IFT | LARGE |
|---|---|---|---|
| Falcon | 0.59±0.04 | 0.93±0.26 | 0.65±0.00 |
| Llama 2 | 0.34±0.02 | 0.58±0.00 | 0.36±0.05 |
| Codellama | 0.62±0.01 | 0.00±0.00 | 0.38±0.02 |
| Mistral | 0.53±0.29 | 0.68±0.00 | 0.66±0.01 |

(d) COMPCOMB

| Model | BASE | IFT | LARGE |
|---|---|---|---|
| Falcon | 0.41 | 0.47 | 0.55 |
| Llama 2 | 0.54 | 0.58 | 0.61 |
| Codellama | 0.46 | 0.49 | 0.60 |
| Mistral | 0.39 | 0.51 | 0.59 |

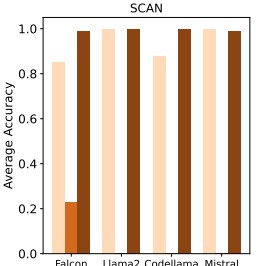 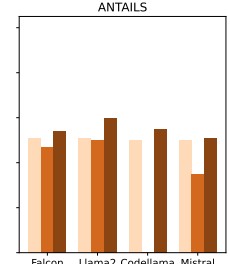 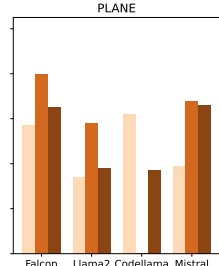 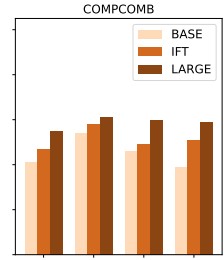

Figure 4: Comparison of average accuracies across different model families (Falcon, LLama, Codellama, Mistral) and model types (BASE, IFT, LARGE) for four datasets (SCAN, Antails, Plane, CompComb). Each bar represents the average accuracy across two prompts in the Prompt setup.

Table 3: SCAN Templates across two setups to extract the comparative log probabilities.

**SETUP 1**

```
'''The command "[command_example1]"
is written as "[action_sequence_example1]".

The command "[command_example2]"
is written as "[action_sequence_example2]".

The command "[command_example3]"
is written as "[action_sequence_example3]".

The command "[command_example4]"
is written as "[action_sequence_example4]".

The command "{command}" is written as
"{true_action}".'''

'''The command "[command_example1]"
is written as "[action_sequence_example1]".

The command "[command_example2]"
is written as "[action_sequence_example2]".

The command "[command_example3]"
is written as "[action_sequence_example3]".

The command "[command_example4]"
is written as "[action_sequence_example4]".

The command "{command}" is written as
"{control_action}".'''
```

**SETUP 2**

```
'''The command "[command_example1]" translates to
"[action_sequence_example1]".

The command "[command_example2]" translates to
"[action_sequence_example2]".

The command "[command_example3]" translates to
"[action_sequence_example3]".

The command "[command_example4]" translates to
"[action_sequence_example4]".

The command "{command}" can be translated to
"{true_action}".'''

'''The command "[command_example1]" translates to
"[action_sequence_example1]".

The command "[command_example2]" translates to
"[action_sequence_example2]".

The command "[command_example3]" translates to
"[action_sequence_example3]".

The command "[command_example4]" translates to
"[action_sequence_example4]".

The command "{command}" can be translated to
"{control_action}".'''
```

Table 4: Antails Templates across two setups to extract the comparative log probabilities.

### SETUP 1

```
'''Here is the premise and the hypothesis:
    Premise: {p}.
    Hypothesis: {h}.
    Question: Determine the entailment relation between the
    premise and the hypothesis.
    Answer: The premise does entail the hypothesis'''

'''Here is the premise and the hypothesis:
    Premise: {p}.
    Hypothesis: {h}.
    Question: Determine the entailment relation between the
    premise and the hypothesis.
    Answer: The premise does not entail the hypothesis'''
```

### SETUP 2

```
'''"{p}" does entail "{h}"'''

'''"{p}" does not entail "{h}"'''
```

Table 5: PLANE Templates across two setups to extract the comparative log probabilities.

### SETUP 1

```
'''"{seq_list[0]}" is {lab_list[0]}."{seq_list[1]}" is {lab_list[1]}.
It is the case that {seq_list[2]}'''

'''"{seq_list[0]}" is {lab_list[0]}."{seq_list[1]}" is {lab_list[1]}.
It is not the case that {seq_list[2]}'''
```

### SETUP 2

```
'''"{seq_list[0]}" is {lab_list[0]}."{seq_list[1]}" is {lab_list[1]}.
It holds true that {seq_list[2]}'''

'''"{seq_list[0]}" is {lab_list[0]}. "{seq_list[1]}" is {lab_list[1]}.
It does not hold true that {seq_list[2]}'''
```

