# OpenReview forum: "Toward a Sheaf-Theoretic Understanding of Compositionality in Large Language Models"
_ICLR.cc/2025/Conference — Submitted to ICLR 2025_

### Official Review · Reviewer_DdYq · 2024-10-24

**Soundness:** 3
**Presentation:** 2
**Contribution:** 3
**Rating:** 5
**Confidence:** 3

**Summary:**

1. This work proposes a novel way to use sheaf theory for compositionality for Large Language Models (LLMs). Compositionality is a central concept of human cognition that understands complex things through simpler components. This new definition of compositionally contains four distinct conditions: restriction maps, gluing conditions, locality conditions, and natural transformations.

2. The experiments conducted across multiple LLMs (such as Llama2, CodeLlama, and Mistral) show that larger models tend to exhibit better compositional abilities overall. However, instruction-tuned models experience a significant decline in performance, particularly in tasks related to the restriction condition and natural transformations. This suggests that while instruction tuning may enhance generalization, it can degrade the model's ability to handle compositional information.

**Strengths:**

1. The paper introduces a completely new, higher-level definition of compositionality using sheaf theory, which provides a fresh perspective on evaluating the compositional abilities of LLMs. This novel approach broadens the understanding of how complex linguistic expressions are structured and processed by LLMs. Provide a more comprehensive way to measure LLMs' ability.

2. The authors evaluate various LLMs (such as Llama2, CodeLlama, and Mistral) across four different angles, providing a detailed comparison of their compositional performance. This analysis helps identify which models excel in certain tasks and where they fall short, offering useful insights for improving future models.

**Weaknesses:**

1. The paper does not include a dedicated Related Work section, which is critical for situating the proposed framework within the existing literature. This omission makes it difficult to understand how the new approach builds upon it.

2. The paper falls short of the 10-page limit. This space can be used to conduct analysis deeper.

3. While the proposed sheaf-theoretic framework is innovative and offers a high-level definition of compositionality, its real-world applicability remains uncertain. The framework may be too abstract or theoretical for immediate use in practical model development or evaluation, raising questions about its tangible impact.

4. The paper does not provide a better solution for improving compositionality in LLMs.

**Questions:**

1. Are there any situations or specific tasks where the proposed sheaf-theoretic framework might not be applicable?

2. The performance shifts observed in Table 1 indicate differences across various models. What insights can these performance variations provide regarding model architecture, training methodologies, or the training data used?

---

> ### Author Response · Authors · 2024-11-23
> **Reply to Reviewer DdYq**
>
> We would like to thank the reviewer for their valuable feedback which has been immensely helpful for us to  revise and update our paper!
>
> With regards to the weaknesses:
>
> Related Work and Page Limit: In the updated draft, we have added a related work section, situating our contributions within the existing literature on compositionality in LLMs. Additionally, we have used the remaining page space to for the new Related Work section and extending our Discussion section as well.
>
> Real-World Applicability: We recognize that the sheaf-theoretic framework may appear abstract, but its purpose is to formalize compositionality into distinct, testable conditions. To bridge theory and practice, we include mappings from each condition to specific datasets in the appendix, illustrating how these formal aspects can be evaluated in real-world LLMs. This approach is particularly relevant to ongoing research into reasoning and generalization, where compositionality plays a foundational role. The Discussion section now also contains more on this.
>
> Improving Compositionality in LLMs: The aim of our work is not to propose methods for improving compositionality but to provide a structured framework for evaluating it. By highlighting compositionality’s multifaceted nature, our framework can help future works assess whether specific techniques improve reasoning and generalization capabilities in LLMs.
>
> With regards to the questions, we have addressed the point on real world applicability above. For insights on performance variations,  we have added clarification in the Limitations section that our dataset samples are quite small and we might need to work on building bigger datasets in the future for this. The observed performance shifts reflect differences in models’ abilities to handle specific compositional tasks and investigations on this will require much more detailed considerations that is beyond the scope of the current one. Future work could extend these insights by applying the framework to a broader range of architectures.

---

> > ### Comment · Reviewer_DdYq · 2024-11-28
> > **Reply to authors**
> >
> > Thanks for your reply, I will keep my score.

---

### Official Review · Reviewer_mmvK · 2024-11-01

**Soundness:** 3
**Presentation:** 2
**Contribution:** 3
**Rating:** 5
**Confidence:** 2

**Summary:**

Compositionality is a topic on human cognition.
LLMs appear to show their superior language processing capability and thus evaluating the compositionality become a portal to gain insights into these models.
This paper introduced a sheaf-theoretic framework with 4 different datasets to assess LLMs performance on compositionality.
Key findings include 1) larger models tend to be more performant. 2) instruction finetuned models may behave inconsistently in different tasks.

**Strengths:**

- Paper investigated both query (aka, prompt output) and internal representation in different tasks.
- To reviewer's knowledge, this paper is clear and original in proposing the sheaf-theoretic framework for LLMs compositionality assessment.
- Two orders of relationship are investigated, entity level and relation level.
- Proposed 4 datasets, namely SCAN, ANTAILS, COMPCOMB, PLANE, aim to unveil insights in restriction maps, gluing condition, locality condition, and natural transformation.
- Appendix provided abundant information about generation process of each dataset.

**Weaknesses:**

- A related work section would help readers to better understand the background and prepare readers well to follow the sheaf-theoretic framework.
- While 4 tasks cover distinct aspects, the size of these dataset could be limited to capture the compositionality of LLMs.
- Compositionality is an important in human cognition and investigating it in LLMs is also exciting. The paper would be more complete if authors include the importance of LLMs compositionality in applications. What are the aspects or benefits if LLMs gains better compositionality performance.
- There is limited description of the experiment setup. It would be preferable if there are more justifications for the methodology and choice of prompting.
- (line 908) There is a table or figure missing about the SCAN setup, showing **??**
- Appendix also shows the template used but it would be better to include additional examples in each dataset to help readers gain insights.
- There are many extremely long sentences, which requires a second pass and thus affect the readability of the paper.

**Questions:**

- Is llama2 chat hf an instruction-following checkpoint?
- Would it be possible to include Qwen in the evaluation for additional trends and patterns revealing? Also llama released 3 and 3.1.
- When introducing 4 tasks, the ordering is not consistent, is there any special consideration?
For example, the order in table 1 or figure 1 (1SCAN, 2ANTAILS, 3COMPCOMB, 4PLANE) is different from the order in section 2 and section 3.3.
- What is the size of each dataset?
- Could you provide an example for each dataset?
- Could chain-of-thought prompting apply during evaluations?

---

> ### Author Response · Authors · 2024-11-23
> **Reply to Reviewer mmvK**
>
> We would like to thank the reviewer for their feedback and for helping us realize the potential shortcomings of our work!  We have updated our draft to address possible shortcomings. With regards to the weaknesses:
>
> Related Work Section: We appreciate this very helpful suggestion and have added a detailed related work section in the updated draft. This section provides background on compositionality in cognition, its applications to LLMs, and recent advances in the field, preparing readers to better understand our framework and why we need it.
>
> Dataset Size and Representation: We acknowledge that the datasets used are limited in size and scope. However, our intention is not to provide exhaustive datasets but to demonstrate that compositionality can and should be evaluated across distinct aspects. We now clarify this point in the Limitations section and encourage further work in creating larger, more diverse datasets for this purpose.
> Importance of Compositionality in Applications: We agree that discussing the benefits of improved compositionality in LLMs would enhance the paper. Compositionality underpins reasoning and learning, making it central to tasks requiring systematic generalization and robust understanding. We have added this in our Discussion section, emphasizing how improved compositionality can advance LLM capabilities in reasoning-based tasks.
>
> Experimental Setup and Justifications: Our evaluation strategy considers the limitations of LLMs, particularly with long outputs (e.g., SCAN) where exact match evaluation is not feasible under standard prompting methods. Advanced techniques like chain-of-thought prompting introduce confounding factors, disproportionately benefiting instruction-tuned models, and were thus avoided to ensure fair comparisons. For yes/no entailment tasks, our setup mitigates biases in model outputs by directly assessing probabilities. We have now clarified these choices in Section 4.2 of the revised draft.
>
> Missing Table: This was due to a missing label tag and we have now addressed this in our revision.
>
> Examples: We have added examples in the appendix explanations  for each dataset to provide clearer insights.
>
> Readability: We have shortened long sentences throughout the paper to improve readability. If there are more specific confusing sentences, we would be happy to improve that in the camera ready.
>
> With regards to the questions,
>
> Models:The llama-chat is an instruction following checkpoint. Our table in the appendix gives links to Huggingface pages for all models we used, and more information can also be found there for each model. With regards to the addition of models, we appreciate the suggestion and plan to include evaluations on these models in future work. Unfortunately, resource constraints prevent us from adding them to the current submission.
>
> Dataset Size and Examples: We have included the size of each dataset now and also representative examples in the appendix for clarity. The ordering of tasks and datasets is not specific and we can change this ordering to make it more consistent for the camera ready.
>
> Chain-of-Thought (CoT) Prompting: CoT prompting was avoided as it disproportionately benefits instruction-tuned models and introduces uncertainties in how it influences model performance. Our evaluation aims to assess intrinsic compositional abilities without introducing such external factors.

---

> > ### Comment · Reviewer_mmvK · 2024-11-28
> >
> > Thank you for clarifying my questions and updating the draft. After reviewing the changes, I decided to keep the original score.

---

### Official Review · Reviewer_SWD5 · 2024-11-04

**Soundness:** 2
**Presentation:** 3
**Contribution:** 1
**Rating:** 3
**Confidence:** 4

**Summary:**

The paper introduces a mathematical framework for compositionality in LLMs based on sheaf topology, defining four basic conditions: restriction maps, gluing conditions, locality conditions, and natural transformations. The authors tested each condition with a specific dataset and found that instruction-tuned models have inconsistent performance across different aspects of compositionality.

**Strengths:**

- The paper presents a nice initial effort in defining compositionality with a mathematical framework rigorously
- The paper is well-structured and well-written

**Weaknesses:**

- (Major) Limited novelty and applicability of the findings:
	- The formalization of compositionality via the sheaf-theoric framework is novel, but it is not clear its purpose, how it can be used in practice in the real world and whether it's correct or not (due to a weak experimental part).
	- There is no novelty in the evaluation part since it uses known datasets and results for the literature, except for the introduction of the COMPCOMB dataset.
	- In general, there are very few contributions that justify this paper.
- (Critical) The experimental part is extremely weak and not convincing:
	- In Table 1 there is a significant drop between base models and instruction-tuned models like a huge ~40% (0.82 vs 0.42) on SCAN. The authors provide an explanation that  L402 "instruction tuning likely leads to a loss in the development of restriction maps, which could be explained by the fact that while the model retains its most important generalizations, it loses some local information to accommodate instruction tuning, leading to loss of restriction mapping". This is in contrast with results in the literature where instruction-tuned models perform generally better. I believe there is not enough empirical evidence to sustain this statement and the huge performance drop might be due to issues in the evaluation setup. The authors mention the use of "computing the model’s log probabilities for two possible completions" but this has been shown to be problematic, especially in instruction-tuned models that might lose the calibration in their logits after RLHF (https://arxiv.org/abs/2402.14499 , https://arxiv.org/abs/2303.08774). I suggest using a different eval strategy (e.g., comparison with the ground truth and exact match metric) and distinguishing results between instruction-tuning and models tuned via RLHF.
	- In general, I don't think the experiments are thorough enough to convince me without any doubt that an increase/decrease of the score on a specific dataset (e.g., SCAN) means an increase/decrease of a compositional condition defined in the framework (e.g., Restriction Condition). There are several other factors involved in the evaluation that might lead to spurious correlations and an increase/decrease in the score. I think the author should definitively pay more attention to the evaluation of the components defined in the framework.
- (Minor) The presentation of the results is not optimal:
	- The results proposed (Figures and Tables) are never referenced from the text. This creates ambiguity in the text because it's not clear what results you are commenting on.
	- The proposed plots lack clarity. The rationale for using a radar plot to compare accuracies in Figure 1 is unclear, and both the scale and raw values in the plot are difficult to interpret.
- (Minor) There is a minimal discussion of related works. Missing related papers (e.g., [A Complexity-Based Theory of Compositionality](https://arxiv.org/abs/2410.14817))

**Questions:**

N/A

---

> ### Author Response · Authors · 2024-11-23
> **Reply to Reviewer SWD5**
>
> We are extremely grateful for the detailed analysis and review of our work and would like to thank the reviewer for their valuable time! On the basis of the comments, we provide the following clarifications:
>
> Purpose and Applicability: The motivation for our work, as stated in the introduction, is to provide a theoretical foundation for evaluating compositionality in LLMs. Current definitions and datasets focus on narrow aspects of compositionality, leaving gaps in comprehensive understanding of a typically symbolic phenomenon for conneectionist systems . Our sheaf-theoretic framework formalizes compositionality as a multi-faceted phenomenon, addressing both local data structure and global relational structure.
> The focus of the work is not to show that we have the best framework or the best tasks for compositionality but that formalizing via our framework allows us to gain a deeper understanding of the phenomenon itself. While no framework can be definitively “correct,” given the lack of ground truth for compositionality in human cognition, our work demonstrates the need to evaluate compositionality through diverse tasks, encouraging the community to pursue more nuanced approaches. We realize that our writing might not have been clear enough on these points and have updated the Abstract , the Introduction and also the Discussion section accordingly.
>
> Novelty: The novelty of our work lies in two main aspects- showing that typically cognitive notions like compositionality are often ill-defined in terms of models like LLMs and lack formalization in terms of what is to be evaluated. Introducing a formal framework for analyzing compositionality across multiple levels that show how formalization can help better define such phenomenon and also come up with more concrete evaluation tasks. While the tasks and datasets used in our experiments are not new, they serve to validate the framework’s applicability rather than claim novelty in experimental design. The framework provides a structured way to analyze compositionality, moving beyond ad hoc approaches, and offers insights into LLM behavior that existing evaluations may overlook. In our new draft, we have also updated the limitations section to clarify this point.
>
> Experimental Results: Our evaluation strategy reflects practical limitations. Exact match evaluation was infeasible for SCAN due to long output strings which models cannot do with simple prompting techniques, and advanced prompting techniques like chain-of-thought introduce confounding factors that can unfairly favor instruction-tuned models. Our log-probability-based approach aimed to provide a consistent comparison, but we recognize its limitations and will explore alternatives in future work.
> Regarding the performance drop in instruction-tuned models, recent studies (e.g., https://arxiv.org/abs/2402.05119) show that instruction tuning can degrade performance in some settings. While our results align with this trend, we agree that more empirical evidence would strengthen this conclusion and will address this in future iterations.
>
> Presentation: We have revised the text to ensure all tables and figures are explicitly referenced. Radar plots were chosen to provide a comprehensive visual comparison of model accuracies across multiple datasets and setups. While we recognize potential clarity issues and are open to simpler visualizations for the camera-ready version, radar plots offer a quick overview. The radial axis, scaled from 0 to 1, highlights top-performing models. For detailed metrics, please refer to the bar plots and tables.
>
> Related Work: The new draft has an expanded discussion of related work to include additional references, such as “A Complexity-Based Theory of Compositionality”, to better contextualize our contributions.
>
> We appreciate the reviewer’s feedback and acknowledge the need for stronger empirical validation and improved presentation. Our primary contribution is the formal framework, which highlights the need for nuanced, multi-faceted evaluation of compositionality in LLMs. We hope our work will serve to inspire awareness of using cognitive theory and formalization to improve the nature of empirical research in this area.

---

> > ### Comment · Reviewer_SWD5 · 2024-11-26
> >
> > I thank the authors for their reply.
> >
> > > While no framework can be definitively “correct,” given the lack of ground truth for compositionality in human cognition, our work demonstrates the need to evaluate compositionality through diverse tasks, encouraging the community to pursue more nuanced approaches.
> >
> > I understand that no framework can be deemed universally "correct." However, the decision to use a particular framework over others should be well-justified and demonstrate clear advantages. One of the major weaknesses here is that this choice appears somewhat arbitrary and does not seem to offer any particularly compelling benefits.
> >
> > > notions like compositionality are often ill-defined in terms of models like LLMs and lack formalization in terms of what is to be evaluated.
> >
> > As per my original review, I don't think that the framework proposed solves this "lack formalization in terms of what is to be evaluated". In general, I doubt that an increase/decrease of the score on a specific dataset (e.g., SCAN) means an increase/decrease of a compositional condition defined in the framework (e.g., Restriction Condition) as several factors in the evaluation that might play a role.
> >
> > I thank the authors once again for their reply. I believe that several major concerns raised in my original review remain unaddressed. I encourage the authors to revise the paper to address these issues. At this time, I feel it is appropriate to maintain my original score.

---

> > > ### Author Response · Authors · 2024-11-27
> > > **Response to Reviewer SWD5**
> > >
> > > Thank you very much for your interest in our work. Please let us know what concerns have not been addressed to provide further clarifications and avoid misunderstandings.
> > >
> > > > the decision to use a particular framework over others should be well-justified
> > >
> > > There are currently no other frameworks modelling compositionality. It is usually considered a linguistic task with a variety of purposes and datasets used. We will make sure to clarify this in the paper. This is the first, to our knowledge, attempt to develop a more nuanced evaluation framework.
> > >
> > > > I doubt that an increase/decrease of the score on a specific dataset (e.g., SCAN) means an increase/decrease of a compositional condition
> > >
> > > We agree with the reviewer and this is the exact reason we suggest a multifaceted sheaf theoretic framework instead of using individual datasets. This framework provides an evaluation that covers multiple aspects of compositionality and can be expanded by modifying datasets to serve the purpose of the tasks we suggest: gluing conditions, restriction maps, locality conditions, and learning universal transformations.

---

> > > > ### Comment · Reviewer_SWD5 · 2024-11-27
> > > >
> > > > > There are currently no other frameworks modelling compositionality.
> > > >
> > > > This is clear. My point is that there are multiple ways to mathematically model the problem of compositionality. While you propose an approach based on sheaf topological spaces, the paper does not provide sufficient evidence to demonstrate why this choice should be preferred over other potential options, nor does it provide evidence that this choice is well-justified and has a clear advantage. In essence, it remains unclear how to use the proposed framework and why it stands out as a compelling choice. This is particularly important given the absence of proof or any definitive means to establish the correctness of the framework.

---

> > > > > ### Author Response · Authors · 2024-11-27
> > > > > **Response to Reviewer SWD5**
> > > > >
> > > > > We want to thank the reviewer for their comment.
> > > > >
> > > > > In our "Introduction", we refer to two papers from theoretical and experimental cognitive science that were instrumental in our choice of the sheaf-theoretic framework:
> > > > >
> > > > > - Phillips (2018):  https://www.frontiersin.org/journals/psychology/articles/10.3389/fpsyg.2018.01926/full
> > > > > - Phillips (2020):  https://royalsocietypublishing.org/doi/full/10.1098/rstb.2019.0303
> > > > >
> > > > > These papers emphasize the utility and uniqueness of the sheaf-theoretic framework as a method for modelling compositionality and how this mathematical framework not only explains compositional generalization but can also explain human cognition with respect to behaviour in generalization. The works however focus on human cognition only and our work uses the grounding provided in these works to develop our framework for large language models.
> > > > >
> > > > > If needed, we would be willing to expand our introduction or add a section in the appendix detailing the motivation from these papers to better explain why the sheaf theoretic approach is perhaps the only rigorous way of modelling compositionality that also explains the property.

---

> > > > > > ### Comment · Reviewer_SWD5 · 2024-11-27
> > > > > >
> > > > > > I thank the authors once again for their additional comments.
> > > > > > At this time, I feel it is appropriate to maintain my original score.

---

### Comment · Area_Chair_oKeR · 2024-11-25
**To reviewers: please discuss**

The rebuttal was submitted late in the discussion period, so there isn't much time left to discuss with them directly. If they have addressed your concerns or if you have further questions, please let them know.

---

### Author Response · Authors · 2024-12-04
**Official Closing Comment from Authors**

We would like to sincerely thank the reviewers for their detailed feedback and engagement, which has significantly improved the quality and clarity of our work. Below is a summary of the changes made in response to critiques and comments.

## Purpose and Scope Clarifications
- In light of the comments from all reviewers, we felt that one of the primary issues was the lack of clarity behind the need for our proposed approach to studying compositionality.
- We have now revised the **Abstract** and **Introduction** sections to clarify the lack of any existing framework for studying an important cognitive phenomenon like compositionality and thus a need for an attempt at formalizing compositionality. We now explicitly highlight in our work that our primary contribution is providing a first-of-its-kind theoretical framework for evaluating compositionality in LLMs, rather than claiming experimental or dataset novelty. The paper now explicitly emphasizes the importance of formalizing compositionality as a multi-faceted phenomenon and explains how our framework addresses both local and global structures. We elaborated on how the framework formalizes typically ill-defined cognitive notions like compositionality and provides structured evaluations that complement existing, often ad hoc, approaches.
- We have also added a new **Related Work** section to contextualize our contributions in the literature better, referencing works like *“A Complexity-Based Theory of Compositionality.”*
- We enhanced the **Discussion** section to elaborate on the practical implications of improved compositionality, particularly for reasoning-based tasks and systematic generalization.
- The **Limitations** section now highlights that our datasets and tasks are validation tools for the framework rather than claims of experimental innovation.

## Experimental Setup and Results
- **Evaluation Justifications:** We clarified why certain evaluation strategies (e.g., avoiding CoT prompting and using log probabilities) were chosen, discussing their strengths and limitations in **Section 4.2**.
- **Performance Trends:** Added references to recent studies (e.g., instruction-tuning effects) to contextualize observed performance drops in instruction-tuned models and acknowledge the need for further empirical validation in future work.
- **Dataset Size and Examples:** We included dataset sizes and representative examples in the appendix, improving transparency and readability.

## Presentation Improvements
- All tables and figures are now explicitly referenced, and a missing table label was corrected.
- Examples of dataset tasks were added in the appendix, and the text was revised to improve readability by shortening long sentences.
- We retained radar plots for comparative summaries but are open to simpler visualizations for the camera-ready version.

## Additional Clarifications
- **Model Usage and Links:** We clarified model details and included relevant links in the appendix. While resource constraints limit the inclusion of additional models, we plan to extend this in future work.

We appreciate your valuable feedback, which has helped us refine and improve our paper. Our primary goal remains to provide a formal framework that advances understanding and evaluation of compositionality in LLMs, encouraging more nuanced approaches in this area. We hope these revisions meet your expectations and address your concerns.

Thank you once again for your time and insightful comments!

---

### Meta-Review · Area_Chair_oKeR · 2024-12-04

**Metareview:**

This paper proposes a new sheaf theoretic framework for reasoning about compositional generalization. They use this framework to define a taxonomy of different facets of compositionality.

The presentation is clear and well structured; the authors have made further presentation improvements during the discussion phase. Reviewers praised the originality of the chief theoretic framework for compositionality.

However, reviewers remained unconvinced about the advantages of this theoretical framework over alternative ways of thinking about compositionality. They were unconvinced by both the correctness and by the possibility of new applications of this theory. The paper would be improved by showing how sheaf theory can be used to make novel predictions or explain phenomena around compositionality better than alternative theoretical models like Bayesian reasoning about possible automata, or better than alternative taxonomies of compositionality such as Hupkes et al. 2020.

**Additional Comments On Reviewer Discussion:**

The authors made some presentation improvements and clarifications, but did not manage to address the main criticisms in the reviews. All reviewers responded by maintaining their scores.

---

### Decision · Program_Chairs · 2025-01-22

Reject